# Use of the Milan Pet Quality of Life Instrument (MPQL) to Measure Pets’ Quality of Life during COVID-19

**DOI:** 10.3390/ani11051336

**Published:** 2021-05-08

**Authors:** Patrizia Piotti, Christos Karagiannis, Liam Satchell, Manuela Michelazzi, Mariangela Albertini, Enrico Alleva, Federica Pirrone

**Affiliations:** 1Department of Veterinary Medicine, University of Milan, 26900 Lodi, Italy; patrizia.piotti1@unimi.it (P.P.); Federica.pirrone@unimi.it (F.P.); 2Hellenic Institute of Canine and Feline Behaviour & Training, 10434 Athens, Greece; ckaragiannis@behaviour.gr; 3Department of Psychology, University of Winchester, Winchester SO22 4NR, UK; Liam.Satchell@winchester.ac.uk; 4Parco Canile Rifugio, 20134 Milan, Italy; manu.michelazzi@gmail.com; 5Centro di Riferimento per le Scienze Comportamentali e la Salute Mentale (SCIC)—ISS, 00161 Rome, Italy; Enrico.alleva@iss.it

**Keywords:** quality of life, personality, One Health, COVID-19, dog, cat

## Abstract

**Simple Summary:**

COVID-19 caused unprecedented lifestyle changes, with unknown effects on pets’ quality of life. We studied the role of personality, the human–animal relationship, COVID-19-related factors, and demographics on feline and canine quality of life (QoL). We used the novel Milan Pet Quality of Life instrument (MPQL), derived from previous scales, which summarises four QoL domains: physical, psychological, social, and environmental. Our findings indicate that pets’ demographics, life experience, and personality could explain a large part of the physical QoL. Conversely, the owners’ demographics, COVID-19-related changes, and the owners’ personality mostly explained the pets’ psychological QoL. Predictably, outdoor access in the home environment and the country of residence mostly explained the pets’ environmental QoL. Finally, the pets’ social QoL was explained by all previous aspects: pets’ characteristics and personality, environment and COVID-19-related changes, and the pet–human relationship. We suggest that these results may be explained by the effects of the COVID-19 pandemic on the owners’ psychological state and lifestyle, which in turn alter the way the owners interact with their pets and look after them. However, the owners’ personality and mood may also influence the way they interpret and report their pets’ behavior and emotional state. These findings highlight the importance of the One Health–One Welfare perspective.

**Abstract:**

The COVID-19 pandemic caused lifestyle changes, with unknown effect on pets’ quality of life (QoL). Between May and July 2020, we distributed an online survey to investigate the role of several factors on feline and canine QoL, including lockdown-related factors. We used existing scales to measure human and pets’ personalities (Reinforcement Sensitivity Theory Personality Questionnaire, RST-PQ; RST-Dog; RST-Cat) and the human–animal relationship (Lexington Attachment to Pets Scale, LAPS) and the Milan Pet Quality of Life instrument (MPQL). Overall, 235 participants reported about 242 adult pets (N_cats_ = 78, N_dogs_ = 164). Factor analysis confirmed the structure and internal reliability of the existing scales (RST-PQ, RST-Dog, RST-Cat, LAPS) and suggested a four-factor structure for the MPQL (physical, psychological, social, environmental). The results indicate that the pets’ psysical QoL was largely explained by pet-related elements (pets’ demographics and life experience, and pets’ personality). Conversely, the pets’ psychological QoL was explained mostly by owner-related elements, such as the owners’ demographics, COVID-19-related changes, and the owners’ personality. Predictably, the pets’ environmental QoL is mostly explained by environmental factors, such as the outdoor access in the home environment and the country. Finally, the pets’ social QoL was explained by the larger combination of models: pets’ characteristics and personality, environment and COVID-19-related changes, and the pet–human relationship. These findings can be explained by two non-mutually exclusive mechanisms. The reported changes may be a by-product of the COVID-19 pandemic’s psychological and lifestyle effects on the owners, which in turn alter the way the owners interact with their pets and look after them. However, the owners’ characteristics and mood may bias their answers regarding their pets.

## 1. Introduction

The pandemic caused by coronavirus disease 2019 (COVID-19) led to unprecedented global measures aimed to reduce social contact and population movement to contain the spread of the virus. However, the same measures have been shown to affect the psychological wellbeing of the general population [1]. Similarly, COVID-19 restrictions are expected to have an impact on the behaviour and welfare of non-human animals [2]. Some authors suggest that less human disturbance in the environment may be of great benefit to some animal species, such as wildlife [3], while other species could be adversely affected. For example, companion animals might develop separation-related issues linked to the constant presence of their owners at home, their health might be affected by the postponement of non-urgent veterinary interventions adopted by veterinary practices in several countries, and, finally, owners might have less funds to spend on health care for their pets [2]. Recent findings indicate that companion animals showed increasing signs of stress due to the social and environmental changes that occurred during periods of partial or total lockdown [3]. Thus, from the *One Health–One Welfare* perspective, the pet–human relationship under COVID-19 should be investigated in depth [4].

There is a large body of research investigating the beneficial role of pet ownership on human wellbeing. Domestic animals are often considered by their owners as family members and perceived as having an important role in one’s wellbeing [5,6,7]. There is even evidence of the fact that the presence of pets has mitigated some of the detrimental psychological effects that COVID-19 lockdown has on humans’ quality of life (QoL) [8]. QoL is reflective of human and other animals’ wellbeing, which can be evaluated through psychometric scales and observed in the behaviour of pets [9]. For the purpose of this work, QoL is defined as “*an individual’s satisfaction with its physical and psychological health, its physical and social environment, and its ability to interact with that environment*”, as suggested by Belshaw and colleagues [10]. Additionally, the World Health Organization (WHO) states that QoL models should: measure life experiences relevant to the individual; reflect the variety of life aspects; consider contexts beyond physical welfare; measure experiences at population and individual level [11,12]. The assessment of QoL should measure experiences at an individual level, however, research on the role of personality on individuals’ QoL is currently underrepresented. Personality differences predispose to problematic behaviour in humans and non-human animals. In humans, strong tendencies to seek reinforcers predispose to impulsive and risk-taking behaviour [13,14], while tendencies for repetitive and fearful thoughts predispose to develop generalised anxiety, depression, and phobias [13]. However, in humans, fear was also found be functional during the COVID-19 lockdown, increasing public health compliance [15].

Conversely, the literature regarding the *effects* of *pet ownership* on pets has highlighted several critical aspects. Owners’ previous experience and attitudes when acquiring a dog, training habits, consistency, level of engagement with the dog, and management choices are all associated with the development of behaviour problems in the animals [16,17]. The literature also indicates that cat owners’ personalities and the access to hiding and climbing spots can affect animal welfare [18,19,20], although cats’ psychological welfare and the role of the social environment are largely understudied [20,21]. Additionally, the owners’ age, gender, and family composition influence cats’ QoL [22]. In dogs, QoL is positively affected by their bond with their owners and negatively affected by the pets’ age [23,24]. This further supports the need for adopting a *One Health–One Welfare* perspective and assessing pets’ QoL during the COVID-19 pandemic. Veterinary sciences have similarly recognised the multidimensionality of QoL [9,10,25]. Various tools are present in the scientific literature to describe canine QoL (see [10] for a review). However, only two of them covered all of the WHO domains [23,26]. One of these tools [26] was developed as a transcript for over-the-phone interviews rather than a self-reporting questionnaire. In the other scale [23], items covered only aspects of the pet’s life relative to negative affect, trainability, and fulfilment of basic physiological needs. For this reason, we used a novel QoL instrument, the Milan Pet Quality of Life (MPQL) tool, on the basis of existing literature. Given the link between fearful and aggressive personality traits and avoidance behaviour in dogs [27], and the relationship between dogs and owners’ personality and the development of behaviour problems [28], we argue that, when looking at quality of life, it is important to take into account both the pet and the owner’s personality in a way that favours cross-species comparison.

Human and non-human animals’ personalities can be characterised through reinforcement sensitivity theory (RST), a neuropsychological approach for the understanding of individual differences [13,29]. It describes the underlying psychological and biological processes concerned with *approaching* and *avoiding* appetitive and aversive stimuli [30]. The RST of personality is constructed around dispositional inclination to engage in three neuropsychological systems. The behavioural approach system (BAS) is sensitive to rewarding stimuli and has the role of moving the individual towards a biological reinforcer, within a temporo-spatial dimension [30]. The flight–fight–freeze System (FFFS) is a punishment sensitivity system that mediates reactions to all aversive stimuli, which initiates avoidance behaviours, including defensive aggression and freezing [31,32]. Finally, the behavioural inhibition system (BIS) is sensitive to goal conflict (e.g., equal activation of the BAS and FFFS towards incompatible or ambiguous approach and avoidance goals, respectively). The behavioural outputs of the BIS evolved to allow the individual to enter a potentially dangerous situation (i.e., leading to cautious approach behaviour) or to withhold entrance (i.e., initiates passive avoidance) if the conflict is sufficiently intense [30]. Human predisposition for the activation of the RST subdomains can be measured through a psychometric scale for adult participants, such as the RST Personality Questionnaire (RST-PQ) [30]. Recently, a scale for the measurement of RST subdomains in dogs was developed to be completed by their owner or caregiver (RSTPQ-Dog) [27].

Previous research looking into the effect of COVID-19 lockdown on pets took into account owners’ lifestyle changes and the pet–owner relationship but not pets and owners’ personalities [3]. Therefore, the current study aimed to expand on previous research and measure the effect of pets and humans’ personality traits on pets’ QoL during the COVID-19 pandemic, in addition to the previously investigated factors, such as pets and humans’ demographic and environmental factors, lockdown-related factors, and the pet–human relationship. In relation to this relationship, we hypothesised that high scores in the human and pet RST domains would be detrimental to pets’ QoL by causing impulsive, fearful, and/or anxious responses to changes. As previously observed, we also expected the pet–owner relationship, demographics, and factors related to COVID-19 lockdown to affect the pets’ QoL.

## 2. Materials and Methods

The study was approved by the Ethics Committee (CE) of the University of Milan (n. 39/20 dated 30/04/2020). Participants were given written information about the aim and the procedures of the study and the right to withdraw at any time. In addition, they were informed that the survey was anonymous, and confidentiality would be maintained by the researchers. Before data collection, informed consent was obtained from each participant. Participation was voluntary.

### 2.1. Participants

Participants of this study were recruited through convenience sampling using social media and word of mouth.

#### 2.1.1. Inclusion Criteria

Participants were required to be above the age of 18. If they were pet owners, they were also required to own only cats and/or dogs (no other concomitant species of domestic or exotic animals). This was decided in order to avoid the confounding factors linked to the relationship with other pets. They were also required to have lived with their pet(s) for at least 6 months. They could report for as many pets as they wished.

#### 2.1.2. Exclusion Criteria

During data collection, the survey ended if the participant responded that they owned other pets than cats and/or dogs. Questionnaires were excluded from the study if they were not completed in full. Additionally, completed surveys were excluded from further analysis if the owner had been living with the pet for less than 6 months.

### 2.2. Survey

A custom-made online survey was produced for this study, uploaded on the online sampling platform Qualtrics XM, both in the English and Italian languages, and distributed between 5 May 2020 and 8 July 2020, a period when several countries globally were affected by various degrees of lockdown measures.

The survey was produced by a multidisciplinary team of academics and experts from the public health sector. The questionnaire included both validated scales and new items generated through expert consensus aimed at measuring financial, psychological, social, and environmental aspects related to the COVID-19 pandemic and relative lockdown measures. For some questionnaires (see below), validated Italian versions exist, and for the others, the original English version was translated into Italian by a bilingual researcher (PP). It was then back translated into English by a bilingual researcher, and the two versions were compared and corrected by two bilingual researchers, one native in Italian (PP) and the other native in English. The final questionnaire consisted of the following 8 sections:

#### 2.2.1. Demographic Data about the Participant

The demographic information obtained included the participants’ age, gender, nationality and state of residence at the time of participation, household composition, relationship status and co-living status, level of education, geographical characteristics of their living area, type of residence, how the responder perceived the size of their home, and whether they had unrestricted access to outdoor space. Details about the demographic questions can be found in Appendix A (Table A1).

#### 2.2.2. Questions Relative to the COVID-19 Pandemic

Responders were asked how they perceived their risk of contracting COVID-19 (high, medium, low) and whether they were currently ill with COVID-19-related and/or unrelated conditions. Since the questionnaire was distributed internationally and given that, at the time, the actions taken by various governments varied frequently, participants were enquired about the restrictions imposed by their country at the time of participation. Responders were asked to report about the severity of the restriction measures issued by their country at the time of participation (national lockdown, social distancing, or no limitations) and how much their freedom of movement and engagement in social interactions had been restricted in the past 15 days.

Since financial factors might have a great impact on the level of distress of the population during the pandemic, responders were also asked to report they were facing financial issues due to the COVID-19 pandemic.

Finally, responders were asked whether they judged their current wellbeing overall to be better, the same, or worsened, compared to prior to the pandemic.

#### 2.2.3. Human Personality (RST-Human)

This section included a validated scale of human personality, based on reinforcement sensitivity theory, a psychobiological construct that summarises behavioural tendencies to approach and avoid stimuli, and motivational conflict [30]. This questionnaire was translated into Italian and back translated. The questionnaire consists of 65 items on a 4-point Likert scale and describes the three RST domains, BAS, BIS, FFFS. Additionally, the scale provides four subdomains for the BAS domain. These are: BAS-Impulsivity, i.e., a tendency towards unplanned behaviour; BAS-Reward Reactivity, i.e., high sensitivity to rewards; BAS-Goal Drive Persistence, i.e., a tendency to pursue future rewards; BAS-Reward Interest, i.e., high reward-seeking behaviour. A high score indicates high sensitivity to and/or activation of that domain or subdomain, showing that the related trait is very strong in that individual, while a low score indicates low sensitivity and/or activation.

#### 2.2.4. Pet Demographics

For each pet they reported for, responders were asked to indicate whether it was a cat or a dog. They were then asked the age of their pet, length of the ownership, sex and neutering status of the pet, whether it was a cross or pure breed, and size. Since it is well known that early experiences have an important role in the development of behaviour problems [33,34], we enquired in detail about these aspects. The demographic questions are described in Appendix B (Table A1).

#### 2.2.5. Personality of the Pet (RST-Pet)

The pets’ personalities were measured using a reinforcement sensitivity theory scale, originally developed in English for dogs [27]. The questionnaire consisted of 21 items on a 5-point Likert scale, which described the three RST domains, BAS, BIS, and FFFS. As in the human version, a high score indicates high sensitivity to and/or activation of that domain or subdomain, while a low score indicates low sensitivity and/or activation. This questionnaire was translated into Italian and back translated.

#### 2.2.6. Quality of Life of the Pet (Milan Pet Quality of Life, MPQL)

The MPQL was derived from two validated scales for the measurement of non-health-related quality of life of dogs [23,26]. Some questions were adapted from the original telephone transcript [26] for use as an online survey and to ensure that all questions were on a 4-point Likert scale. Furthermore, a number of questions were added based on experts’ agreement and a previous scale [23], in order to measure the four QoL domains identified by the WHO, i.e., physical health, psychological health, social satisfaction and support, and environmental safety and satisfaction. Specifically, questions about signs of organic disease and behavioural signs of physical ailments were added. Questions about freedom of movement and safety in the living area, as well as training activities, were rephrased so that there were questions directed to cat owners and questions directed to dog owners. The first draft of the questionnaire was piloted with a small sample of pet owners (N = 10), who were invited to indicate whether the questionnaire was clear and suggest changes or missing questions. This questionnaire was produced both in English and Italian at the same time. The two versions were then compared by two bilingual researchers, one native in English and the other (PP) native in Italian.

#### 2.2.7. Pet–Owner Relationship (LAPS)

The pet–owner relationship was measured using the LAPS validated scale [35] which describes the relationship according to 3 domains: a general attachment domain, a people substitute domain, indicating that the pet in question occupies a more central position in the respondent’s life, and an animal welfare/animal rights domain. The scale questionnaire consists of 23 items on a 4-point Likert scale. A high score in a domain indicates the owner’s relationship with their pet is close to what is described in that domain. The validated Italian version of the scale was also used [36].

### 2.3. Statistical Analysis

Analyses were carried out using R statistical software [37]. The packages psych [38], GPArotation [39], and lavaan [40] were used for factor analyses and the package lme4 [41,42] for regressions.

Some of the levels of certain explanatory variables were merged for statistical analysis. Specifically, the relationship status was arranged in 3 categories (in a relationship and co-living, in a relationship and not co-living, not in a relationship), as we predicted that, among various relationship statuses, the fact of having an affective relationship and the fact of sharing a home environment during lockdown were the aspects that most likely would affect human QoL. We also summarised level of education into two groups (up to secondary education and tertiary education and above) as there is a cross-cultural relationship between educational level and economic inequality [43]. Some of the questions related to the living situation were also simplified to reduce the complexity of the models: the perceived size of homes was transformed into a binary variable (small vs. medium/large) as those perceiving their home as being small were expected to be most affected by lockdown measures; for the type of private outdoor space, garden and fields were aggregated, providing a relatively similar experience compared to balconies or no outdoor space at all. Finally, we summarised the levels of some of the questions related to perceived issues related to the COVID-19 pandemic, as our variables of interest are binary in nature, but multiple options were given to responders for the survey to be easier to understand. Specifically, the variables that we transformed into binary categories were the perceived risk of contracting COVID-19 (no risk vs. medium/high), illness of the participant (yes/no), limits on movement (no/minimum limitations vs. extensive limitations), financial loss (none/minimum financial distress vs. significant financial distress).

Further, some variables pertinent to the pet demographics were categorised into groups. Breed type was separated in pure breeds with a pedigree vs. mix and pure breeds without a pedigree (for the impossibility of confirming the genealogy of the pet without a pedigree). The age at adoption was categorised based on whether they were adopted earlier than 8 weeks (as earlier separation from the mother and the litter are associated with the development of behaviour problems in dogs and cats [33,44,45]), between 8 and 13 weeks, or above 3 months of age (which is commonly considered a late adoption). The origins of the pet were classified based on factors predictive of the development of behaviour problems and poor socialisation, such as unregistered breeders and pet shops [33], those for whom the early experiences are generally unknown, such as shelters or pets found in the streets, and those typically associated with known and controlled early experiences, such as private individuals, friends, or regulated breeders. Finally, we categorised the reasons for the adoption of the pet. Specifically, answers were summarised into four categories: emotional reasons (company, family member, rescue, pet in need for a home), work (guard, removing pests, protection, police, hunting, herding, sport), support to the self, and support to others; however, when looking at the descriptive statistics, it appeared that the variables lacked variance, so they have not been included in the following analysis.

We began with explanatory factor analysis (EFA) of the MPQL in order to examine the factor structure of the novel measure, and to identify any redundant items. Twenty-six items were entered into this EFA. The results of parallel analysis and the exploration of the scree plot suggested four factors should be extracted. Therefore, on this basis, four factors were extracted with oblimin rotation and a factor loading cut-off below 0.2. Six items were removed for insufficient loading (reproductive issues, time alone, interaction with strangers, familiar people and other animals, indoor access) and one item (frequency of walks) was removed for cross loading (factor loadings are reported in Appendix B).

Subsequently, we explored item response distributions of the RST questionnaires (RST-Human, RST-Pet) that had not been previously validated in Italian and we performed a confirmatory factor analysis (CFA), hypothesising that items should group into the same construct as in the original version of each questionnaire. The internal consistency (reliability) of all the construct scales was also evaluated by Cronbach’s alpha.

We then investigated the relationship between demographics, COVID-19-related factors, personality, and pet–owner relationship on the domains of pets’ QoL as measured by the MPQL. The data were analysed using a series of linear regression models which were calculated for each MPQL domain, including the following fixed factors: (1) a pet demographics model including the species, age, sex and neutering status, breed, size, body condition score; (2) a life experience model including the origin of the pet, the age at adoption, and the length of the pet ownership; (3) an owner demographics model including level of education, age, gender, relationship status, presence of current medical conditions; (4) an environment model including the geographycal area, the type of home, the size of the home, access to outdoor areas, and country of residence; (5) a COVID-19 model including the owner’s perceived risk of contracting COVID-19, the official lockdown situation, the level of limitations to movement, financial loss; (6) a human RST model including the RST domains FFFS, BIS, BAS-Impulsivity, BAS-Reward Reactivity, BAS-Goal Drive Persistence, BAS-Reward Interest from the human RST questionnaire; (7) a pet RST model including the RST domains FFFS, BIS, BAS from the pet RST questionnaire; (8) a pet–human relationship model including the LAPS domains general attachment, people substitute, animal rights. Each regression included the pet as a random factor. In order to evaluate the power of explanation of each model, we calculated Nagelkerke’s R squared (pseudo-R squared) and calculated the model fit by comparison via an ANOVA with the null model. Finally, pairwise post hoc comparisons with Tukey correction were then obtained for categorical fixed factors.

## 3. Results

### 3.1. Subjects

Overall, 628 participants engaged with the study link. However, 240 records were excluded from subsequent analysis because the survey had not been filled in fully, while 115 declared they had other pets, and therefore the survey was automatically aborted by the system; another seven records were excluded because the responder declared they had owned the animal for less than 6 months. The final sample consisted of 291 records, of these, 235 included pet reports, including 161 dogs and 74 cats. Among pet owners, 19 reported about more than one pet (two to five pets).

The final sample consisted mostly of female responders (85%, N = 248). The majority of responders were Italians (79%, N = 227) aged between 25 and 54 years (75%, N = 217). Consequently, they mostly responded from Italy (79%, N = 222) and chose Italian as their language (81%, N = 237). Mostly, responders were either in a co-living relationship (54%, N = 156) or not in a relationship at all (33%, N = 96) and they had a tertiary level degree or above (79%, N = 231). Demographic data are reported in Appendix A (Table A1).

### 3.2. Descriptive Results

The majority of responders declared that they lived in an urban or suburban area (76%, N = 221), as opposed to responders living in a rural area (24%, N = 70). Most of the responders deemed their home to be of medium or large size (82%, N = 240), mostly with a balcony (47%, N = 137) or a garden and fields (41%, N = 119). In general, the majority of the sampled population had some sort of unrestricted access to the outdoors (62%, N = 183).

The perceived risk of contracting COVID-19 reported by the responders was similarly distributed between low and medium–high perceived risk (low: 44%, N = 102; medium–high: 56%, N = 133) and the majority of responders reported that they did not suffer from medical conditions (84%, N = 243), related or unrelated to COVID-19.

The majority of responders also reported that, at the time of participation, their country of residency had issued some level of social distancing measures (75%, N = 219), rather than complete lockdown (23%, N = 66) or no measures at all (2%, N = 6). The majority of responders also reported no or slight financial distress caused by the pandemic (81%, N = 236).

Finally, the majority of responders reported either worse or same level of wellbeing compared to prior to the pandemic (worse: 40%, N = 116; same: 45%, N = 131). The results of the descriptive analysis are reported in Appendix A (Table A1).

### 3.3. Pets’ Characteristics

Of the 235 pets, the majority were neutered females (44%, N = 104), aged 1–5 years (37%, N = 88) and 5–10 years (38%, N = 90). The pets were similarly distributed in terms of pure breeds vs. mixed breeds, size, and age at adoption (Table 1). The majority of responders had been living with the pet for 1–5 years (43%, N = 102) and 5–10 years (33%, N = 78) and there was an equal distribution in terms of sources of the dog (pet shop or unregistered breeder, private owner or registered breeder, shelters, and found in the street, Table 1).

When enquiring about the attitudes around adopting the pet (Table 1), we found that the majority of the owners reported emotional reasons (98%, N = 231). Only a few responders reported working reasons (17%, N = 39), and support to the self (9%, N = 22) or others (3%, N = 8). Demographic questions about the pet are reported in Appendix B.

### 3.4. Factor Analysis and Questionnaire Validation

CFAs confirmed the structure of the Italian version of the RST-PQ (χ^2^ (df = 145) = 4491.47, *p* < 0.001, CFI = 0.86, TLI = 0.85, RMSEA = 0.07) and the RST-Pet (χ^2^ (df = 45) = 296.88, *p* < 0.001, CFI = 0.96, TLI = 0.96, RMSEA = 0.06) scales. Furthermore, Cronbach’s alpha demonstrated the Italian versions of the RST-PQ scale to be reliable (BAS-Interest α = 0.77; BAS-Persistence α = 0.81; BAS-Reward Reactivity α = 0.80; BAS-Impulsivity α = 0.70; BIS α = 0.93; FFFS α = 0.76) and the Italian version of the RST-Pet scale to be highly reliable for all domains (BAS α = 0.86; BIS α = 0.89; FFFS α = 0.82).

To evaluate the factor solution of the MPQL, several well-recognised criteria for the factorability of a correlation were used. Firstly, it was observed that 24 of the 26 items correlated by at least 0.2 with at least one other item, suggesting reasonable factorability. Secondly, the Kaiser–Meyer–Olkin measure of sampling adequacy was 0.62, above the commonly recommended value of 0.60, Bartlett’s test of sphericity was significant (χ^2^ (325) = 853.77, *p* < 0.001), and the determinant of the correlation matrix was 0.02. Given these overall indicators, factor analysis was deemed to be suitable with all 26 items. Exploratory factor analysis (EFA) was used to discover the factor structure of the MPQL. Parallel analysis (PA) suggested a four-factor solution. Initial eigenvalues indicated that the first four factors explained 30%, 22%, 21%, and 19% of the variance, respectively. The four-factor solution, suggested by PA, which explained 19% of the variance, was preferred because of: (a) its theoretical support [11]; (b) the scree plot; and (c) the insufficient number of primary loadings and difficulty of interpreting additional factors. As some overlapping between factors was expected, the oblimin rotation was used for the final solution. A total of seven items were eliminated because they did not contribute to a simple factor structure and failed to meet a minimum criteria of having a primary factor loading of 0.2 or above, and no cross loading of 0.2 or above (see Appendix B). For the final stage, a principal component factor analysis of the remaining 21 items, using varimax and oblimin rotations, was conducted, with three factors explaining 18% of the variance. An oblimin rotation provided the best defined factor structure. All items in this analysis had primary loadings over 0.2. The factor loading matrix for this final solution is presented in Appendix B (Table A2).

The factor labels proposed by the WHO [11] suited the extracted factors and were retained. Factor 1 of the MPQL scale was composed of items reflecting the facet of physical QoL, such as signs of pain and reduced mobility, sensory decline, dependence on medication, and amount of play. Factor 2 was composed of items regarding signs of negative affect, use of punishment by the owner, and frequency of procedures that the pet perceives as unpleasant (such trimming nails or cleaning ears), thus reflecting the facet of psychological QoL. Factor 3 was composed of items reflecting the facet of environmental QoL, including items related to the pet’s freedom and safety of movement in the environment. Finally, Factor 4 was composed of items reflecting the facet of social QoL, such as the time spent together, frequency of training activities and signs of positive affect in social contexts. For each case, we retained the participant-level residuals of each factor for further analysis. These were reversed for physical, social, and environmental QoL so that for all factors, a higher score indicated a better quality of life and a lower score indicated a poorer quality of life.

#### Pets’ Quality of Life during COVID-19

Physical QoL was significantly explained by models built using the pets’ demographic factors (33% explained variance), the pets’ life experiences (22%), and the pets’ personalities (19%) (Table 2). Post hoc pairwise analysis indicated that within the pets’ demographics, species, age, and breed had a large effect on the score of physical QoL. Particularly, cats had a better QoL than dogs and older pets had a poorer QoL than middle-aged and younger pets. Additionally, the physical QoL appeared to be worse in mixed breeds compared to purebred pets. Looking at the pets’ life experiences, only the length of ownership and the age of adoption had an effect. Specifically, QoL declined with the increasing length of ownership (Length: β = −0.44, *p* < 0.001), while it was better when the pet had been adopted at an appropriate age (around 2 months for dogs and 3 months for cats) rather than a late adoption (Adequate vs. Late: β = −0.30, *p* = 0.031). Regarding the pet’s personality, it appeared that physical QoL declined as the pet’s FFFS and BIS scores increased (FFFS: β = −0.14, *p* = 0.022, BIS: β = −0.17, *p* = 0.003), while it improved with higher pet BAS scores (BAS: β = 0.29, *p* < 0.001). Finally, owner-related factors had a very small effect on physical QoL. The fit of the owner’s characterstics model demonstrated a trend towards significance (*p = 0.049*) and explained only 7% of the variance, while the pet–human relationship explained only 5% of the variance (Table 2). Within the owner’s demographics, the owner’s medical conditions and perceived overall wellbeing had a significant effect on physical QoL. Specifically, owners who reported no medical conditions also reported poorer scores in the physical QoL of their pets (Yes vs. No: β = −0.38, *p* = 0.012). Similarly, owners who perceived their wellbeing to be worse during the lockdown reported poorer scores for their pet’s physical QoL (Same vs. Declined: β = −0.30, *p* = 0.042).

The variance of Psychological QoL was explained mostly by owner-related models (Table 2): the owners’ demographics (12%), COVID-19 related factors (10%), and owners’ personality (10%). The owners’ level of education and medical conditions had significant effects on pets’ psychological QoL. Specifically, owners with higher education reported better psychological QoL of their pet, compared with owners with lower levels of education (Higher vs. Primary/Secondary: β = 0.29, *p* = 0.017), while owners who reported medical conditions also reported poorer psychological QoL in their pet (Yes vs. No: β = −0.30, *p* = 0.017). When looking at the model for COVID-19-related factors, it appeared that the only factor affecting pets’ psychological QoL was the owner’s financial loss, with pets having a better psychological QoL when owners reported small or no financial losses compared to a large loss (Large vs. Small/No Loss: β = −0.58, *p* < 0.001). Finally, in relation to personality, owners reported slightly poorer psychological QoL in their pet with increasing BIS (Human BIS: β = −0.02, *p* < 0.001) and BAS-Reward Interest (Human BAS-Reward Interest: β = −0.04, *p* = 0.021).

The pet-related model for pets’ demographics explained 7% of the psychological QoL’s variance (Table 2) and this was primarily the species of the pet, with cats being associated with a higher score for the psychological QoL (Cat vs. Dog: β = 0.41, *p* = 0.019).

The social QoL factor’s variance was explained by several models (Table 2): the pet personality (30%), pets’ demographics (24%), the environment (20%), COVID-19-related factors (10%), the pet–human relationship (10%), and the pets’ life experiences (8%). Regarding the pet’s personality, the psychological QoL was mostly affected by the BAS, improving significantly with higher pet BAS scores (BAS: β = 0.42, *p* < 0.001), while it declined as the pet’s FFFS increased (FFFS: β = −0.10, *p* = 0.036); the BIS did not appear to have a significant effect (BIS: β = −0.06, *p* = 0.194). In relation to the pets’ demographics, only an effect of breed type was observed, with purebred pets having a better social QoL than mixed breeds. In relation to the environment, access to outdoor space within the household had a significant effect on pets’ social QoL. Specifically, access to a garden or a private field was associated with better social QoL in the pet, compared with houses with just a balcony (Balcony vs. Garden: β = −0.33, *p* = 0.004) or windows (Windows vs. Garden: β = −0.38, *p* = 0.044). Moreover, pets abroad had a higher score for social QoL compared to pets in Italy (Italy vs. Other Countries: β = −0.53, *p* < 0.001). In the COVID-19-related model, several factors affected the pets’ social QoL. Specifically, the pets’ social QoL was better when the owner reported lower perceived risk of contracting COVID-19 (Low vs. Medium/High Risk: β = 0.24, *p* = 0.011) and when the owner encountered minimum limitations (Large vs. No/Minimum Limitations: β = −0.36, *p* = 0.001). However, the pets’ social QoL was better when social distancing measures were in place, compared to no measures (No Measures vs. Social Distancing: β = −0.78, *p* = 0.024). When looking at the pet–owner relationship, only general attachment had a significant effect on the social QoL, which improved slightly with increased reported general attachment to the owner (General Attachment: β = 0.05, *p* = 0.001). Finally, in relation to the pets’ life experiences, only the length of the ownership had a significant, and negative, effect on the social QoL (Length of Ownership: β = −0.17, *p* = 0.002).

For the environmental QoL, only the environment (20%) and the owners’ demographics (8%) models significantly explained a proportion of the variance of the pets’ environmental QoL. Contrary to what observed for the social QoL, access to a garden or a private field was also associated with poorer environmental QoL in the pet, compared with houses with just a balcony (Balcony vs. Garden: β = 0.32, *p* = 0.012) or windows (Windows vs. Garden: β = 0.69, *p* < 0.001). However, similarly to what was seen in the social QoL, pets abroad had a higher score for environmental QoL compared to pets in Italy (Italy vs. Other Countries: β = −0.39, *p* = 0.002). In terms of owners’ demographics, only the gender factor affected the environmental QoL, with female owners reporting a better environmental QoL in their pet compared to male owners (Female vs. Male: β = 0.47, *p* = 0.021).

## 4. Discussion

The main findings of the study indicate that different domains of the pets’ MPQL are differently explained by the investigated models. Specifically, the pets’ physical QoL is largely explained by pet-related elements (pets’ demographics and life experience, and pets’ personality). Conversely, the pets’ psychological QoL is surprisingly explained mostly by owner-related elements, such as the owners’ demographics, COVID-19-related changes, and the owners’ personalities. Predictably, the pets’ environmental QoL is mostly explained by environmental elements, such as the outdoor access in the home environment and the country. Finally, the pets’ social QoL is the domain explained by the larger combination of models: pets’ characteristics and personalities on one side, environment and COVID-19-related changes on the other side, as well as the pet–human relationship. Overall, these findings can be explained by two non-mutually exclusive mechanisms. On one hand, the owners’ characteristics may be biasing their answers regarding their pets. On the other hand, many of the reported changes may be an indirect by-product of the psychological and lifestyle effects that COVID-19 has on the owners, which in turn alter the way the owners interact with their pets and look after them.

With regard to our hypothesis on the role of personality on pets’ QoL, this was partly supported. As predicted, as the pet’s sensitivity to the flight–fight–freeze system (RST-FFFS) and to the behavioural inhibition system (RST-BIS) subdomains increased, their physical and psychological QoL declined. High sensitivity to BIS predicts individual predisposition for anxiety [46], while high sensitivity to FFFS predicts individual predisposition for avoidance behaviours in dogs [27]. Our findings may indicate that FFFS affects the expression of physical discomfort-related negative affect in cats and dogs, as has been observed in humans [47]. We expected that high sensitivity to the behavioural approach system (RST-BAS) subdomain in the owner would predict lower QoL in the pets, however, we observed the opposite effect. The RST-BAS subdomain was associated with better physical and social QoL in pets. In previous research, the BAS was positively correlated with impulsivity measures [27], thus suggesting that higher BAS might lead to more impulsive pets. However, evidence indicates that dogs with a high tendency to approach novel stimuli are more outgoing and less prone to developing medical conditions [48], supporting the current findings that high-BAS pets are indeed more likely to have better physical QoL. The current findings highlight how the BAS measure also represents a positive predisposition in the pet (a positive bias [49]) which may lead to lower expression of physical and emotional discomfort. It should be noted that it was not possible to assess certain aspects of pets’ behaviour using the MPQL, such as the quality of social interactions with humans and pets, as the relative items did not load in the EFA. At the time of data collection, these aspects were likely widely influenced by the lockdown and social distancing measures faced by the majority of responders. Similar findings could be observed in relation to the owners’ personalities: increased RST-BIS sensitivity of the owner predicted poorer physical QoL. It is known that more empathetic dog owners might be prone to noticing more subtle signs of distress in their pet when completing psychometric scales [50], suggesting that, in the current population, owners with high BIS scores might more anxious and therefore more attentive to negative details [51] when describing their pets’ physical health or that they were more affected by the COVID-19 pandemic and differences in their lifestyle, mood, and overall ability to cope with the situation might have affected their ability to look after their pet at this time. Interestingly, the social QoL domain was negatively affected by the owner’s sensitivity to the BAS subfacet “Reward Reactivity”. Reward reactivity reflects an owner who is quick to respond to excitement and experience positive emotions [30]. According to the classical model of extraversion, such an individual would be particularly affected by a decline in the opportunities for seeking novelties and would be quicker to become bored, thus, their behaviour might possibly lead to poorer pet–owner interactions.

Previous research has observed a relationship between pets’ welfare and the pet–owner relationship [3]. Our findings suggest that such a relationship partly explains the social aspects of pets’ QoL and, to a smaller degree, their physical QoL. Specifically, it appears that the “general attachment” described in the LAPS is the drive for more frequent and fulfilling pet–owner interactions. This in turn might indirectly affect the pets’ health, because pets that are more involved in training activities may be more active and healthier, or owners spending more time with their pet may more promptly notice early signs of disease.

Our hypothesis regarding the negative impact of the COVID-19 lockdown on pets’ QoL was partly supported, and it is interesting to notice that COVID-19 lockdown-related factors explained to some degree the psychological and social QoL of pets. This again indicates how the effects of COVID-19 may be an indirect result of a decline in the owners’ mental health and lifestyle changes. When owners reported greater financial loss caused by the lockdown, their pet had a significantly poorer psychological QoL. It has been observed that job insecurity and financial concerns due to COVID-19 are related with depressive symptoms, after accounting for demographics characteristics, health status, and other COVID-19 experiences [1,52]. On one side, owners’ reports of their pets’ psychological QoL may be negatively biased by their own poor mental health. However, dogs are also able to discriminate emotional facial expressions [53] and tend towards avoidance when humans have negative or neutral facial expressions [54]. In addition, inconsistency in owner behaviour negatively affects dogs’ welfare [17]. This could also be the case for cats, although, to our knowledge, the same phenomenon has not been investigated in cats yet, so this could also be the case for this species. Overall, it is reasonable to suppose that inconsistencies caused by poor mental health of their owners may impact the psychological QoL of cats and dogs. As the effect may reflect a negative psychological state of their owners, this is something that governments should consider for the future. Conversely, social QoL appears to be more directly affected by the limitations to movement, this being requested by the government (lockdown), or secondary to the owner’s concern about the risks for their health. However, nearly all responders reported facing either social distancing measures or a full lockdown due to COVID-19. For this reason, the MPQL could not fully assess the pets’ interaction with strangers and visitors, the average amount of time spent alone by the pets, and the frequency and duration of walks.

The results of the study also highlight that the demographic characteristics of the pet have a long-term effect on their physical QoL. Specifically, pets adopted later in life seem to have more health problems. This may be due to older pets being overrepresented in this group. The other demographic aspects that had a large impact on QoL were the pets’ species, their age, and length of ownership. Unsurprisingly, age was a negative predictor for physical QoL. Older pets encounter a decline in their sensory abilities [55,56,57,58,59,60] and they are prone to developing medical conditions, such as osteoarthrosis, diabetes, or cancer [61,62], which affect their QoL [63]. It is also interesting to notice that the length of ownership appears to have a negative impact on both the physical and social QoL of the pets. We believe that this result is simply a proxy for the pets’ age—as the length of ownership is likely closely related with the age of the pet. According to this view, as the pet grows older, physical conditions increase and the amount, frequency, and intensity of the interactions with owners decrease [64]. In relation to species, according to our results, cats had higher scores than dogs in the physical QoL and lower scores in the social QoL. We believe that cats’ better physical QoL is an artefact caused by cats’ subtle expression of physical discomfort. Evidence indicates that cats communicate their experience of pain mostly through facial expressions [65,66], hiding, and immobility [67], which may be difficult to interpret for pet owners. On the other hand, cats’ lower scores of social QoL likely represents the owners’ limited engagement in joint activities, such as training or exercise, and lack of interaction with the pet, as cat owners may be prone to believing that cats are self-regulated in terms of exercise and prefer solo over social play [68].

Finally, we found differences in the social and environmental QoL across countries. As these domains included items regarding the management of the pets, such as arrangements when left alone, use of kennels, access to the outdoors, and free roaming, variance in the scores likely reflects different practices among countries.

## 5. Conclusions

Overall, the results of this study indicate that the pets’ QoL during COVID-19 was predominantly affected by pet-related factors when looking at physical QoL, while psychological and social QoL were indirectly affected by the psychological and physical state of their owner. Finally, environmental QoL, which relates to safety and freedom to access the environment, was the domain most directly affected by the COVID-19 pandemic.

The current study also highlights the complexity with which multiple factors affect a multidimensional concept such as QoL.

The MPQL, derived from previous scales [23,26], is an example of how such factors should be taken into account in order to avoid loss of information. Particularly, the owners’ personalities might largely influence the owners’ responses to psychometric scales, while the pets’ personalities may have an important impact on the pets’ expressions of their physical and emotional wellbeing. Taken together, these findings strengthen recent indications that, like in humans, pets’ behaviour problems are complex, multifactorial, and can interfere with the individual’s ability to function within the social domain and environment [69]. This stresses once more the need for a *One Health* perspective in the assessment of pets’ wellbeing [4,70].

## Figures and Tables

**Table 1 animals-11-01336-t001:** Pets’ characteristics.

	Group		
	Cats % (N)	Dogs % (N)	Total % (N)
**Demographic characteristics of the pet**			
Age of the pet			
6–12 months	3 (2)	3 (5)	3 (7)
1–5 years	35 (26)	38 (62)	37 (88)
5–10 years	35 (26)	40 (64)	38 (90)
More than 10 years	27 (20)	19 (30)	21 (50)
**Sex of the pet**			
Male intact	-	21 (34)	14 (34)
Female intact	-	14 (23)	10 (23)
Male neutered	54 (40)	21 (34)	31 (74)
Female neutered	46 (34)	44 (70)	44 (104)
**Breed**			
Pure breed (with pedigree)	16 (12)	49 (79)	39 (91)
Mix or unregistered breed	84 (62)	51 (82)	61 (144)
**Size (dogs only)**			
Below 10 kg	-	22 (36)	-
10–25 kg	-	45 (73)	-
Above 25 kg	-	32 (52)	-
**Age when adopted**			
Early (before 8 weeks)	24 (18)	40 (65)	43 (101)
Adequate (8–13 weeks)	27 (20)	42 (68)	37 (88)
Late (above 3 months)	49 (36)	17 (28)	20 (46)
**Time since adoption**			
6–12 months	4 (3)	8 (13)	7 (16)
1–5 years	43 (32)	43 (70)	43 (102)
5–10 years	31 (23)	34 (55)	33 (78)
More than 10 years	22 (16)	14 (23)	17 (39)
**Origin of the animal**			
Pet shop/unregistered breeder	15 (11)	24 (38)	21 (49)
Private owner or registered breeder	31 (23)	42 (67)	38 (90)
Shelter/streets	54 (40)	35 (56)	41 (96)
**Attitude of the owner towards the adoption of the pet**			
Emotional			
Yes	100 (74)	98 (157)	98 (231)
No	-	2 (4)	2 (4)
Working			
Yes	14 (10)	16 (26)	15 (36)
No	86 (64)	84 (135)	85 (199)
Support to myself			
Yes	8 (6)	10 (16	9 (22)
No	92 (68)	90 (145)	91 (213)
Support to others			
Yes	3 (2)	4 (6)	3 (8)
No	97 (72)	96 (155)	97 (227)

**Table 2 animals-11-01336-t002:** Results of regression analyses on the MPQL. Linear models estimates (S.E.), *p*. A positive estimate indicates a better QoL in the first term of the comparison, a negative estimate indicates a better QoL in the second term of the comparison. Models which had a significant fit are coloured in grey.

Predictors	Physical QoL	Psychological QoL	Social QoL	Environmental QoL
**Pet’s demographics**	***R_p_*^2^ = 0.33 *p <* 0.001**	***R_p_*^2^ = 0.07 *p* = 0.042**	***R_p_*^2^ = 0.24 *p <* 0.001**	***R_p_*^2^ = 0.03 *p* = 0.542**
Species				
Cat (1) vs. Dog (2)	0.63 (0.16) *<* 0.001	0.41 (0.17) 0.019	−0.14 (0.14) 0.310	0.04 (0.18) 0.823
Pet’s age				
Old (1) vs. Middle age (2)	−0.97 (0.12) *<* 0.001	−0.11 (0.13) 0.653	−0.01 (0.10) 0.992	0.16 (0.13) 0.414
Old (1) vs. Young (2)	−1.14 (0.30) 0.001	−0.24 (0.32) 0.741	−0.29 (0.27) 0.531	0.09 (0.33) 0.960
Young (1) Middle age (2)	0.17 (0.29) 0.829	0.12 (0.31) 0.911	0.27 (0.26) 0.525	0.08 (0.3) 0.968
Pet’s sex				
Female (1) vs. Male (2)	−0.01 (0.09) 0.966	0.05 (0.10) 0.617	0.02 (0.09) 0.854	−0.10 (0.11) 0.337
Breed				
Mix (1) vs. Purebred (2)	−0.31 (0.11) 0.003	−0.07 (0.11) 0.555	−0.33 (0.09) *<* 0.001	−0.08 (0.11) 0.503
Size				
Small (1) vs. Medium (2)	−0.23 (0.15) 0.274	0.07 (0.16) 0.903	−0.31 (0.13) 0.059	−0.16 (0.16) 0.590
Small (1) vs. Large (2)	−0.34 (0.16) 0.098	−0.21 (0.17) 0.452	−0.28 (0.14) 0.139	0.1 (0.18) 0.799
Medium (1) vs. Large (2)	−0.10 (0.13) 0.729	−0.28 (0.14) 0.126	0.04 (0.12) 0.951	0.27 (0.14) 0.142
Body condition score	−0.12 (0.10) 0.251	0.03 (0.11) 0.784	−0.17 (0.09) 0.069	0.01 (0.11) 0.910
**Pet’s life experience**	***R_p_*^2^ = 0.22 *p* < 0.001**	***R_p_*^2^ = 0.04 *p* = 0.091**	***R_p_*^2^ = 0.08 *p* = 0.003**	***R_p_*^2^ = 0.03 *p* = 0.268**
Origin of the pet				
Pet shop/Unregistered breeder (1) vs. Private owner/Registered breeder (2)	−0.08 (0.14) 0.811	0.23 (0.14) 0.244	−0.24 (0.13) 0.134	−0.16 (0.14) 0.514
Pet shop/Unregistered breeder (1) vs. Streets/Shelter (2)	0.13 (0.14) 0.630	0.01 (0.14) 0.997	−0.04 (0.13) 0.951	−0.12 (0.14) 0.680
Private owner/Registered breeder (1) vs. Streets/Shelter (2)	0.21 (0.11) 0.158	−0.22 (0.12) 0.157	0.20 (0.10) 0.130	0.04 (0.12) 0.947
Age at adoption				
Adequate (1) vs. Early (2)	0.02 (0.14) 0.984	0.27 (0.14) 0.149	0.21 (0.13) 0.216	−0.18 (0.14) 0.431
Adequate (1) vs. Late (2)	0.30 (0.11) 0.031	−0.01 (0.12) 0.999	0.13 (0.11) 0.401	−0.25 (0.12) 0.103
Early (1) vs. Late (2)	0.27 (0.14) 0.127	−0.27 (0.144) 0.138	−0.07 (0.13) 0.816	−0.07 (0.14) 0.887
Length pet ownership	−0.44 (0.06) < 0.001	0.055 (0.063) 0.378	−0.17 (0.06) 0.002	0.04 (0.06) 0.490
**Owner’s demographics**	***R_p_*^2^ = 0.07 *p* = 0.049**	***R_p_*^2^ = 0.12 *p* < 0.001**	***R_p_*^2^ = 0.03 *p* = 0.609**	***R_p_*^2^ = 0.08 *p* = 0.019**
Age	0.01 (0.047) 0.820	0.07 (0.04) 0.088	0.01 (0.04) 0.722	−0.05 (0.04) 0.240
Gender				
Female (1) vs. Male (2)	0.13 (0.19) 0.749	−0.04 (0.17) 0.966	0.10 (0.16) 0.827	0.47 (0.17) 0.021
Female (1) vs. Other (2)	0.53 (0.61) 0.659	0.04 (0.55) 0.996	0.48 (0.52) 0.629	−0.59 (0.56) 0.543
Male (1) vs. Other (2)	0.39 (0.63) 0.808	0.08 (0.57) 0.987	0.39 (0.54) 0.757	−1.06 (0.58) 0.163
Level of education				
Higher (1) vs. Primary/Secondary (2)	0.15 (0.13) 0.250	0.29 (0.12) 0.017	0.08 (0.12) 0.503	0.12 (0.13) 0.325
Relationship status				
Co-living relationship (1) vs. No relationship (2)	0.01 (0.12) 0.999	0.09 (0.11) 0.699	−0.20 (0.11) 0.144	−0.06 (0.11) 0.832
Co-living relationship (1) vs. Non-co-living relationship (2)	0.33 (0.18) 0.171	0.14 (0.17) 0.681	0.08 (0.15) 0.858	0.24 (0.17) 0.347
No relationship (1) vs. Non-co-living relationship (2)	0.32 (0.19) 0.211	0.05 (0.18) 0.959	0.289 (0.17) 0.204	0.30 (0.18) 0.211
Medical conditions				
Yes (1) vs. No (2)	−0.38 (0.15) 0.012	−0.3 (0.14) 0.006	0.01 (0.13) 0.963	0.02 (0.14) 0.900
Wellbeing				
Improved (1) vs. Same (2)	0.23 (0.16) 0.335	0.03 (0.15) 0.970	0.06 (0.14) 0.877	0.20 (0.15) 0.382
Improved (1) vs. Declined (2)	−0.06 (0.16) 0.910	0.29 (0.15) 0.128	0.02 (0.14) 0.990	0.29 (0.15) 0.154
Same (1) vs. Declined (2)	−0.30 (0.12) 0.042	0.26 (0.11) 0.055	−0.05 (0.11) 0.892	0.09 (0.11) 0.728
**Environment**	***R_p_*^2^ = 0.01 *p* = 0.853**	***R_p_*^2^ = 0.05 *p* = 0.117**	***R_p_*^2^ = 0.20 *p* < 0.001**	***R_p_*^2^ = 0.20 *p* < 0.001**
Geographical area				
Rural (1) vs. Urban/Suburban (2)	0.15 (0.14) 0.276	−0.02 (0.13) 0.875	−0.05 (0.11) 0.644	−0.08 (0.12) 0.511
Outdoor space				
Windows (1) vs. Balcony (2)	0.02 (0.20) 0.991	0.20 (0.19) 0.519	−0.049 (0.16) 0.948	0.37 (0.17) 0.088
Windows (1) vs. Garden (2)	0.01 (0.20) 0.999	0.17 (0.18) 0.637	−0.38 (0.16) 0.044	0.69 (0.17) < 0.001
Balcony (1) vs. Garden (2)	−0.01 (0.134) 0.988	−0.04 (0.12) 0.949	−0.33 (0.10) 0.004	0.32 (0.11) 0.012
Perceived home size				
Medium/Large (1) vs. Small (2)	−0.06 (0.17) 0.711	0.41 (0.16) 0.009	0.01 (0.13) 0.953	−0.27 (0.14) 0.064
Access to outdoors				
No outdoor access (1) vs. Unrestricted outdoor access (2)	0.15 (0.14) 0.286	0.04 (0.13) 0.749	−0.08 (0.11) 0.445	0.06 (0.12) 0.623
Country of residence				
Italy (1) vs. Other (2)	−0.10 (0.14) 0.486	−0.23 (0.13) 0.081	−0.53 (0.11) < 0.001	−0.39 (0.12) 0.002
**COVID-19**	***R_p_*^2^ = 0.01 *p* = 0.646**	***R_p_*^2^ = 0.10 *p* = 0.001**	***R_p_*^2^ = 0.10 *p* < 0.001**	***R_p_*^2^ = 0.03 *p* = 0.268**
Perceived risk				
Low risk (1) vs. Medium/High risk (2)	−0.01 (0.11) 0.932	0.14 (0.10) 0.167	0.24 (0.09) 0.011	0.09 (0.11) 0.378
Lockdown level				
National/Strict lockdown (1) vs. No measures (2)	0.07 (0.38) 0.979	−0.11 (0.34) 0.942	0.67 (0.31) 0.082	0.21 (0.35) 0.811
National/Strict lockdown (1) vs. Social distancing/Relaxed measures (2)	0.19 (0.14) 0.337	0.12 (0.12) 0.626	−0.11 (0.11) 0.585	−0.15 (0.13) 0.461
No measures (1) vs. Social distancing/Relaxed measures (2)	0.12 (0.36) 0.938	0.23 (0.32) 0.761	−0.78 (0.30) 0.024	−0.37 (0.33) 0.510
Lockdown limitations				
Large limitations (1) vs. No/minimum limitations (2)	−0.12 (0.13) 0.371	0.07 (0.12) 0.551	−0.36 (0.11) 0.001	0.20 (0.12) 0.098
Financial loss				
Large loss (1) vs. No/Small loss (2)	−0.09 (0.14) 0.512	−0.58 (0.13) < 0.001	0.09 (0.11) *0.461*	−0.04 (0.13) 0.763
**Human personality**	***R_p_*^2^ = 0.04 *p* = 0.172**	***R_p_*^2^ = 0.10 *p* < 0.001**	***R_p_*^2^ = 0.02 *p* = 0.242**	***R_p_*^2^ = 0.03 *p* = 0.268**
FFFS	0.01 (0.01) 0.840	−0.01 (0.01) 0.516	0.01 (0.01) 0.918	−0.01 (0.01) 0.911
BIS	−0.01 (0.01) 0.076	−0.02 (0.01) *<* 0.001	0.01 (0.01) 0.798	0.01 (0.01) 0.471
BAS-Impulsivity	0.02 (0.01) 0.274	0.02 (0.01) 0.277	−0.01 (0.01) 0.983	−0.01 (0.01) 0.328
BAS-Reward Reactivity	0.01 (0.01) 0.632	−0.01 (0.01) 0.548	−0.03 (0.01) 0.043	0.02 (0.01) 0.256
BAS-Goal Drive Persistence	−0.04 (0.02) 0.035	0.01 (0.02) 0.678	−0.01 (0.02) 0.362	0.02 (0.02) 0.309
BAS-Reward Interest	0.01 (0.02) 0.602	−0.04 (0.02) 0.021	0.02 (0.01) 0.200	−0.03 (0.02) 0.039
**Pet personality**	***R_p_*^2^ = 0.19 *p* < 0.001**	***R_p_*^2^ = 0.02 *p* = 0.145**	***R_p_*^2^*=* 0.30 *p* < 0.001**	***R_p_*^2^ = 0.01 *p* = 0.481**
FFFS	−0.14 (0.06) 0.022	−0.05 (0.06) 0.396	−0.10 (0.05) 0.036	−0.05 (0.06) 0.452
BIS	−0.17 (0.06) 0.003	−0.03 (0.06) 0.571	−0.06 (0.04) 0.194	−0.02 (0.06) 0.721
BAS	0.29 (0.06) < 0.001	−0.15 (0.067) 0.030	0.42 (0.05) <0.001	−0.09 (0.07) 0.163
**Pet–human relationship**	***R_p_*^2^ = 0.05 *p* = 0.013**	***R_p_*^2^ = 0.01 *p* = 0.632**	***R_p_*^2^*=* 0.10 *p* < 0.001**	***R_p_*^2^ = 0.01 *p* = 0.513**
General attachment	0.06 (0.02) 0.003	−0.01 (0.02) 0.500	0.05 (0.02) 0.001	−0.015 (0.02) 0.403
People substitute	−0.01 (0.02) 0.442	−0.01 (0.02) 0.569	0.02 (0.01) 0.399	0.02 (0.02) 0.161
Animal rights	−0.03 (0.04) 0.469	0.02 (0.04) 0.670	−0.03 (0.03) 0.305	0.01 (0.04) 0.858

Note: FFFS: Flight–Fight–Freeze System. BIS: Behavioural Inhibition System. BAS: Behavioural Approach System.

## Data Availability

The data presented in this study are available on request from the corresponding author. The data are not publicly available due to privacy reasons.

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
