# Peer review of "Use of the Milan Pet Quality of Life Instrument (MPQL) to Measure Pets’ Quality of Life during COVID-19"

_animals, 2021, doi:10.3390/ani11051336_

Round 1

Reviewer 1 Report

General comment – The topic covered by this study has major importance, considering the raised risks of welfare problems in companion animals during the global COVID 19 pandemic. Thus, I recommend the publication of this study after review. My comments and suggestions to improve the text are below. My main concerns are related to the extensive use of abbreviations, the presentation of some results, and the lack of discussion for an important topic of results (3.4. Questionnaires validation).

L33 - Define the abbreviations before their first use.

L34 – Include the number of participants.

L37 – What do ‘pet trait RST-FFFS’ and ‘RST-BIS’ mean? You did not define these traits before. The sentence is impossible to understand without a proper definition of the meaning of these traits.

L38 and 39 – “high owners’ RST-BIS” and “high pets’ RST-BAS”? The same comment in L37.

L40 – “Finally, cats,” – are you referring to all of the cats (as a species) or only for cats characterized for having inconspicuous manifestations of discomfort and considered a ‘low maintenance pet?

L41 – What did you mean as a “…‘low maintenance’ pet”?

L44 - Keywords:Quality of life’ and ‘COVID19’ are already in the title.

L64-65: This sentence lacks a reference.

L83 – 84: Pets predisposed to what? Did you mean pets predisposed to behavioral problems? I could not understand this sentence because the scale should be applied to any individual, including those predisposed or not to certain behavioral manifestations.

L103 – 104: This sentence lacks a reference.

L 246 - Table B1 does not show demographic questions. The table displays the results of the MPQL’s EFA and factor loadings.

L273 – 274: Considering your aims (development and validation of a Quality of Life scale), I suggest including the MPQL questionnaire in Italian and English as a supplementary file.

L339: Why did you consider such a low value (0.2)?

362 - L371: It seems that the fixed factors used are not independent. For example, aren’t type of home and outdoor access correlated? How did you deal with collinearity?

L364: How did you include country of residence as a fixed factor? Some of the countries had very small samples.

L385 – 386: Is this sample size enough comparing to previous studies similar to yours? To me, it seems a very low sample size for a worldwide sampling.

L391 – Could the English version of the MPQL questionnaire be validated based on only 54 responses?

L432 – 433: “Demographic questions about the pet are reported in the Appendix B”. This sentence citing the description of the questions should be in the Material and Methods section since it is not a result.

L441 and forward: There are so many abbreviations making it difficult to follow the text.

L465 – 466: Table B1 shows the factor loadings, not descriptive statistics, as you said.

L451 – 465: Why did you report part of the results for the two versions of the questionnaire separated (L460 - 464), and others results are written in a general way (for both versions together), such as the factor analysis and Cronbach’s alpha results.

Table B1 – I suggest including the eigenvalues and % of variance explained by each factor.

Table 2 – The abbreviations used in the table must be defined in the Tables legend or as a footnote.

Discussion – I suggest avoiding using abbreviations in the discussion to make the text easier to follow.

L501 and forward – not all the readers are familiarized with the personality scales used. So I recommend making a more detailed description of the meaning of the lower or higher values in the scales. For example, what does mean higher values in the ‘pet’s sensitivity to the RST-FFFS’ and ‘RST-BIS subdomains’ (L502 - 503)? In L512, what does mean a high ‘sensitivity to the RST-BAS subdomain’?

L547 – 562 – Perhaps the owners’ negative mental states due to their financial losses induced those people to judge their pets as more depressed or sad. Thus, the relations found should be due to a judgment bias by the owners that perceive their animals as having a worse psychological condition.

The discussion was limited to the block of results “3.4.1. Pet Quality of Life”. It should also be interesting to see a discussion regarding the item of “3.4. Questionnaires validation”. The results of factor analysis on MPQL data were not discussed. Were the four facets found the same as reported in previous studies in companion animals using QL scales?

Some limitations of this study should be mentioned at the end of the discussion. For example, the low sample size for a large range of geographical regions caused many confounding effects (cultural, economic, and social differences) that could not be controlled and affected the results. Considering the number of countries involved, I would expect a much larger sample size in this study. 

L602 – 603: As mentioned in the previous comment, this idea was not explored in the discussion.

Author Response

Reviewer #1

Open Review

English language and style

( ) Extensive editing of English language and style required  
( ) Moderate English changes required  
( ) English language and style are fine/minor spell check required  
(x) I don't feel qualified to judge about the English language and style 

Yes

Can be improved

Must be improved

Not applicable

Does the introduction provide sufficient background and include all relevant references?

(x)

( )

( )

( )

Is the research design appropriate?

(x)

( )

( )

( )

Are the methods adequately described?

( )

(x)

( )

( )

Are the results clearly presented?

( )

(x)

( )

( )

Are the conclusions supported by the results?

(x)

( )

( )

( )

Comments and Suggestions for Authors

General comment – The topic covered by this study has major importance, considering the raised risks of welfare problems in companion animals during the global COVID 19 pandemic. Thus, I recommend the publication of this study after review. My comments and suggestions to improve the text are below. My main concerns are related to the extensive use of abbreviations, the presentation of some results, and the lack of discussion for an important topic of results (3.4. Questionnaires validation).

L33 - Define the abbreviations before their first use.

We have defined the abbreviations has requested.

L34 – Include the number of participants.

The number of participants has now been included.

L37 – What do ‘pet trait RST-FFFS’ and ‘RST-BIS’ mean? You did not define these traits before. The sentence is impossible to understand without a proper definition of the meaning of these traits.

We agree with the Reviewer, so we have now added all these traits’ meaning.

L38 and 39 – “high owners’ RST-BIS” and “high pets’ RST-BAS”? The same comment in L37.

We have added the definitions as suggested by the Reviewer.

L40 – “Finally, cats,” – are you referring to all of the cats (as a species) or only for cats characterized for having inconspicuous manifestations of discomfort and considered a ‘low maintenance pet?

As per syntax, the subordinate refers to the subject (cats) as a species.

L41 – What did you mean as a “…‘low maintenance’ pet”?

We decided to remove the sentence, that was unclear.

L44 - Keywords: ‘Quality of life’ and ‘COVID19’ are already in the title.

As per previous research on the same topic (see, for example, Bowen, J., et al. (2020). The effects of the Spanish COVID-19 lockdown on people, their pets, and the human-animal bond. Journal of Veterinary Behavior40, 75-91), we would prefer to repeat these terms as keywords to help indexing and searching.

.

L64-65: This sentence lacks a reference.

This is the citation that the sentence was referenced to García Pinillos, R. One Welfare Impacts of COVID19 - a Summary of Key Highlights within the One Welfare Framework. Appl. Anim. Behav. Sci. 2021, 105262, doi:10.1016/j.applanim.2021.105262).

L83 – 84: Pets predisposed to what? Did you mean pets predisposed to behavioral problems? I could not understand this sentence because the scale should be applied to any individual, including those predisposed or not to certain behavioral manifestations.

We are grateful to the Reviewer for this comment: the term “predisposed” was a typo and has been deleted.

L103 – 104: This sentence lacks a reference.

This is the citation that the sentence was referenced toDodman, N.H.; Brown, D.C.; Serpell, J.A. Associations between Owner Personality and Psychological Status and the Prevalence of Canine Behavior Problems. PLOS ONE 2018, 13, e0192846, doi:10.1371/journal.pone.0192846).

L 246 - Table B1 does not show demographic questions. The table displays the results of the MPQL’s EFA and factor loadings.

We thank the Reviewer for detecting this mistake. It was table A1 and we have corrected it.

L273 – 274: Considering your aims (development and validation of a Quality of Life scale), I suggest including the MPQL questionnaire in Italian and English as a supplementary file.

As done for other similar scales (DogBARQ, FeBARQ, DIAS, PANAS, CFQ, etc) the two versions of our scale will be freely available from the UNIMI institutional website. We have added a reference to this in the data availability statement.

L339: Why did you consider such a low value (0.2)?

There are different consensuses in choosing cut-off scores. We chose this particular value as a common heuristic.

362 - L371: It seems that the fixed factors used are not independent. For example, aren’t type of home and outdoor access correlated? How did you deal with collinearity?

As these variables are non-linearly related they share only limited correlation. Although they share some variance, they are not directly dependent on each other. In addition we run the Bartlett Test of Sphericity, which checks whether the population correlation matrix resembles an identity matrix (meaning that all variables are perfectly independent from each other) and we checked for collinearity calculating the determinant of the R-matrix. Both tests were satisfactory and have now been reported in the results (lines 498-503)

L364: How did you include country of residence as a fixed factor? Some of the countries had very small samples.

This was addressed as suggested by rev #2. Briefly, all foreign Countries were brought together (Italy vs Other countries).

L385 – 386: Is this sample size enough comparing to previous studies similar to yours? To me, it seems a very low sample size for a worldwide sampling.

Although we agree with the Reviewer that this sample size could be considered small in absolute terms, this study only analyses data within its own target population, therefore sample size in relation to the general population does not affect the value of the results.

L391 – Could the English version of the MPQL questionnaire be validated based on only 54 responses?

We have rephrased all sentences regarding validation throughout the paper.

L432 – 433: “Demographic questions about the pet are reported in the Appendix B”. This sentence citing the description of the questions should be in the Material and Methods section since it is not a result.

The sentence actually referred to factor loadings, so the sentencehas now been corrected and we kept it in the Results section.

L441 and forward: There are so many abbreviations making it difficult to follow the text.

We believe that the paragraph has as many abbreviations as the previous one and would be more difficult to follow if all acronyms were spelled out.

L465 – 466: Table B1 shows the factor loadings, not descriptive statistics, as you said.

We have now corrected the Table.

L451 – 465: Why did you report part of the results for the two versions of the questionnaire separated (L460 - 464), and others results are written in a general way (for both versions together), such as the factor analysis and Cronbach’s alpha results.

CFAs are not results themselves, as they are demonstrations of tool structure. The reason why we used them was to confirm the structure of both versions of the tool.

Table B1 – I suggest including the eigenvalues and % of variance explained by each factor.

According to this suggestion, we have added those values (lines 506-508).

Table 2 – The abbreviations used in the table must be defined in the Tables legend or as a footnote.

We have now defined all the abbreviations.

Discussion – I suggest avoiding using abbreviations in the discussion to make the text easier to follow.

We thank the Reviewer for this suggestion. However, in our opinion the text would be more difficult to follow if all acronyms were spelled out.

L501 and forward – not all the readers are familiarized with the personality scales used. So I recommend making a more detailed description of the meaning of the lower or higher values in the scales. For example, what does mean higher values in the ‘pet’s sensitivity to the RST-FFFS’ and ‘RST-BIS subdomains’ (L502 - 503)? In L512, what does mean a high ‘sensitivity to the RST-BAS subdomain’?

This has been now explained more clearly.

L547 – 562 – Perhaps the owners’ negative mental states due to their financial losses induced those people to judge their pets as more depressed or sad. Thus, the relations found should be due to a judgment bias by the owners that perceive their animals as having a worse psychological condition.

We agree with the Reviewer. This has been argued in the discussion section (lines 757-759).

The discussion was limited to the block of results “3.4.1. Pet Quality of Life”. It should also be interesting to see a discussion regarding the item of “3.4. Questionnaires validation”. The results of factor analysis on MPQL data were not discussed. Were the four facets found the same as reported in previous studies in companion animals using QL scales?

A factor analysis itself is not an inferential test. It is not really designed to be analysed and interpreted beyond how the items group in the data. In our case, the results were in line with what would be expected and an extensive discussion of this would not be informative for readers.

Some limitations of this study should be mentioned at the end of the discussion. For example, the low sample size for a large range of geographical regions caused many confounding effects (cultural, economic, and social differences) that could not be controlled and affected the results. Considering the number of countries involved, I would expect a much larger sample size in this study. 

This was addressed as suggested by rev #2 and the other countries have been merged into one level (Other countries), so that this is now a two level analysis (Italy vs Other countries).

L602 – 603: As mentioned in the previous comment, this idea was not explored in the discussion.

We are really sorry but, unfortunately, we were not able to find the match between the line numbers and the concept this comment referred to.

Reviewer 2 Report

The paper reports on a survey held during the peek of the lockdown, on the factors that affect pet quality of life. In general the paper is promising, but I  have a couple of concerns

What are your aims? The introduction and abstract remain a bit ambiguous: is it designing a new questionnaire about QoL? Or is it testing the impact of personality (of owners and pets) on QoL? Or is it assessing the impact of covid on QoL? Please be specific and organize your MS more along these lines. It would be good if one of these is your main aim and the other just secondary but now you don’t seem to choose and that makes it confusing. The introduction (paragraph 2) was also a bit confusing because it swings between the human and the animals back and forth. I think the exposition would benefit if you streamline this a bit.

After reading the paper, I would think that testing the impact of personality (pet and owner) on QoL is the main aim, but then the question becomes what to do with the covid and the questionnaire development? Perhaps you can downplay the development of the questionnaire as you build on existing instruments and you don’t follow established scale development procedures (qulatitative phase generating items, pruning items, and then confirming the selection and validating with other instruments). That the scale doesn’t work well doesn’t help either. But then I wouldn”’t make a big deal of its name. If you want to keep a focus on the covid, the question becomes how to combine this with the personality factor. Wouldn’t it be an interesting question to look at interactions? How do different types of owners and pets (depending on their personality) react to the covid (measures), perhaps as a holistic block. Not sure if this helps. But now you have too many variable,s and it is not clear what you research and not even what you find.

Why is the impact of psychological QoL not discussed in the results section ?

Statistically, I have two concerns

  1. The internal consistency of the QoL scale is low. It is not enough to just mention this and then proceed as if nothing is wrong. Low internal consistency basically means that the average of these items does not mean anything useful. So I would argue that your analysis do not make sense for most of your factors. I would suggest that you further explore the items: could you bring more/ all items together into one factor (awaiting further research about the structure) with sufficient internal consistency? That would also help to improve the readability of your paper (now it is complex, what is your take away message? People have no good idea of your finding beyond that BAS has some positive effect on QoL
    1. If putting everything together doesn’t solve the problem, I think you will have to analyze item by item, put these in appendix, and try to distill some general tendency and report this in the text.
  2. You overanalyse some variables: country of residence is problematic as the other countries beyond Italy are severly underrepresented. I would ignore this variable, and just check if you results are robust if you only use the Italian residents vs the extended dataset. You really can’t do all these comparisons, you are comparing very small groups and these groups have barely anything representative of their country.

Exposition

  • The English is ok but there are several grammatical errots to it would be good to have it edited.
  • Abstract: I Odon’t think you can assume thatall readers know the abbreviations for the qustionnaires. Perhaps you can also omit the details of the specific findings and say that there are some links between reward and punishment strateties among the owners and pets on the one hand, and QoL on the other hand.
  • Can you assume that all readers know ‘the one health one welfare’ perspective? It may be good to define it in one sentence or so.

Good luck !

Author Response

Reviewer #2

Open Review

English language and style

( ) Extensive editing of English language and style required  
(x) Moderate English changes required  
( ) English language and style are fine/minor spell check required  
( ) I don't feel qualified to judge about the English language and style 

Yes

Can be improved

Must be improved

Not applicable

Does the introduction provide sufficient background and include all relevant references?

( )

( )

(x)

( )

Is the research design appropriate?

( )

(x)

( )

( )

Are the methods adequately described?

( )

(x)

( )

( )

Are the results clearly presented?

( )

( )

(x)

( )

Are the conclusions supported by the results?

( )

( )

(x)

( )

Comments and Suggestions for Authors

The paper reports on a survey held during the peek of the lockdown, on the factors that affect pet quality of life. In general the paper is promising, but I have a couple of concerns

What are your aims? The introduction and abstract remain a bit ambiguous: is it designing a new questionnaire about QoL? Or is it testing the impact of personality (of owners and pets) on QoL? Or is it assessing the impact of covid on QoL? Please be specific and organize your MS more along these lines. It would be good if one of these is your main aim and the other just secondary but now you don’t seem to choose and that makes it confusing. The introduction (paragraph 2) was also a bit confusing because it swings between the human and the animals back and forth. I think the exposition would benefit if you streamline this a bit.

We thank the Reviewer for these suggestions. Thus, our introduction has been better organised and we have provided better explanations.

After reading the paper, I would think that testing the impact of personality (pet and owner) on QoL is the main aim, but then the question becomes what to do with the covid and the questionnaire development? Perhaps you can downplay the development of the questionnaire as you build on existing instruments and you don’t follow established scale development procedures (qulatitative phase generating items, pruning items, and then confirming the selection and validating with other instruments). That the scale doesn’t work well doesn’t help either. But then I wouldn”’t make a big deal of its name. If you want to keep a focus on the covid, the question becomes how to combine this with the personality factor. Wouldn’t it be an interesting question to look at interactions? How do different types of owners and pets (depending on their personality) react to the covid (measures), perhaps as a holistic block. Not sure if this helps. But now you have too many variable,s and it is not clear what you research and not even what you find.

As suggested by the reviewer we downplayed the arguments about validation. We think the reviewer raises an interesting question about interactions here. However, we are not particularly confident that adding interactions would benefit this study. Primarily, we note that adding interactions of this type would require a much larger sample size to draw meaningful inference. We would be cautious about drawing inference from questions our study is not powered to answer. Including interactions would also greatly increase the number of tests being run and add complexity to the paper, whilst not adding robust additional findings. The reviewer does present interesting questions for future research however, and it may be interesting to see research on owner x pet personality interactions in studies powered to test that.

Why is the impact of psychological QoL not discussed in the results section ?

Statistically, I have two concerns

  1.  The internal consistency of the QoL scale is low. It is not enough to just mention this and then proceed as if nothing is wrong. Low internal consistency basically means that the average of these items does not mean anything useful. So I would argue that your analysis do not make sense for most of your factors. I would suggest that you further explore the items: could you bring more/ all items together into one factor (awaiting further research about the structure) with sufficient internal consistency? That would also help to improve the readability of your paper (now it is complex, what is your take away message? People have no good idea of your finding beyond that BAS has some positive effect on QoL

We addressed the concerns regarding internal consistency by calculating the regressions using the residuals of the EFA, which are not based on averages and take into account the EFA’s loadings.

Parallel analyses suggested that a four-factor solution best fit the data. We chose to follow this simulation-data-derived suggestion in our data for best explaining variance. It would not be advisable to coerce this structure into a one factor solution (and doing so would not increase internal consistency).

    1. If putting everything together doesn’t solve the problem, I think you will have to analyze item by item, put these in appendix, and try to distill some general tendency and report this in the text.

  1. You overanalyse some variables: country of residence is problematic as the other countries beyond Italy are severly underrepresented. I would ignore this variable, and just check if you results are robust if you only use the Italian residents vs the extended dataset. You really can’t do all these comparisons, you are comparing very small groups and these groups have barely anything representative of their country.

We appreciate the suggestion and have implemented it.

Exposition

  • The English is ok but there are several grammatical errots to it would be good to have it edited.

The manuscript has been reviewed once again by the English native speaker co-author.

  • Abstract: I Odon’t think you can assume thatall readers know the abbreviations for the qustionnaires. Perhaps you can also omit the details of the specific findings and say that there are some links between reward and punishment strateties among the owners and pets on the one hand, and QoL on the other hand.

We have included explanations in the abstract.

  • Can you assume that all readers know ‘the one health one welfare’ perspective? It may be good to define it in one sentence or so.

It is the one main concept in veterinary and human medicine. Since Animals is now associated with the European College of Veterinary Behavioural Medicine and Animal Welfare, we believe that the audience will be well aware of the concept.